# Marine Microbial-Derived Antibiotics and Biosurfactants as Potential New Agents against Catheter-Associated Urinary Tract Infections

**DOI:** 10.3390/md19050255

**Published:** 2021-04-29

**Authors:** Shuai Zhang, Xinjin Liang, Geoffrey Michael Gadd, Qi Zhao

**Affiliations:** 1School of Mechanical and Aerospace Engineering, Queen’s University Belfast, Belfast BT9 5AH, UK; shuai.zhang@qub.ac.uk; 2The Bryden Center, School of Chemical and Chemistry Engineering, Queen’s University Belfast, Belfast BT7 1NN, UK; x.liang@qub.ac.uk; 3School of Life Sciences, University of Dundee, Dundee DD1 5EH, UK; g.m.gadd@dundee.ac.uk; 4School of Science and Engineering, University of Dundee, Dundee DD1 4HN, UK

**Keywords:** marine microorganisms, urinary catheter, antibiofilm, antifouling, coating

## Abstract

Catheter-associated urinary tract infections (CAUTIs) are among the leading nosocomial infections in the world and have led to the extensive study of various strategies to prevent infection. However, despite an abundance of anti-infection materials having been studied over the last forty-five years, only a few types have come into clinical use, providing an insignificant reduction in CAUTIs. In recent decades, marine resources have emerged as an unexplored area of opportunity offering huge potential in discovering novel bioactive materials to combat human diseases. Some of these materials, such as antimicrobial compounds and biosurfactants synthesized by marine microorganisms, exhibit potent antimicrobial, antiadhesive and antibiofilm activity against a broad spectrum of uropathogens (including multidrug-resistant pathogens) that could be potentially used in urinary catheters to eradicate CAUTIs. This paper summarizes information on the most relevant materials that have been obtained from marine-derived microorganisms over the last decade and discusses their potential as new agents against CAUTIs, providing a prospective proposal for researchers.

## 1. Introduction

Urinary catheters are hollow, partially flexible tubes that are designed to drain urine from the bladder. The earliest use of urinary catheters can be traced back to the third century B.C., but the modern indwelling catheter, called the ‘Foley’ catheter, was designed by Frederick B. Foley in the mid-1930s [1]. To date, over 100 million urinary catheters are used worldwide per year since catheterization rates remain high at 20% in non-intensive care units and 61% in intensive care units (ICUs) [2]. Despite the care taken to avoid contamination, catheters are still susceptible to infections as they provide direct access for uropathogens from the outside environment into the urinary tract, impairing local host defence mechanisms of the bladder [3,4]. Opportunistic uropathogens (Table 1) are mainly faecal or skin microbiota from the patients that can enter the bladder through the catheter lumen (34%) or along the catheter–urethral interface (66%) causing infections and complications, such as encrustation, bladder stones, bacteriuria, pyelonephritis, septicaemia and endotoxic shock (Figure 1) [3,5,6,7]. According to the European Centre for Disease Prevention and Control (ECDC), catheter-associated urinary tract infections (CAUTIs) account for 27% of all hospital-acquired infections in developed countries, and over 1 million cases occur in the USA and Europe [1,8]. In the UK, CAUTIs cost the NHS GBP 1–2.5 billion and account for approximately 2100 deaths annually [9]. 

Depending on clinical indications, the duration of catheterization may be short- (<7 days) or long-term (>28 days). Early work showed that 10–50% of patients undergoing short-term catheterization developed bacteriuria, and all patients undergoing long-term catheterization became infected, regardless of whether the catheter system was open or closed [10]. As a basic survival strategy, bacteria that encounter a catheter surface submerged in urine become attached within minutes [11]. These attached bacteria begin to phenotypically change, producing extracellular polymeric substances (EPS) (mainly exopolysaccharides and proteins) that allow the emerging biofilm community to develop a complex, three-dimensional structure within hours. Once established, the biofilms are very difficult to eradicate and exhibit a high tolerance to antibiotics and other biocidal treatments, making them a continuous focus for infections that can only be eliminated by the constant removal of the catheters [3,7,12,13,14,15]. Furthermore, in the presence of urease-positive bacteria (e.g., *Proteus mirabilis*), urease catalyses the hydrolysis of urea into carbon dioxide and ammonia, which increases the urine pH, leading to the precipitation of calcium and magnesium phosphate crystals (encrustation) and the formation of crystalline biofilms on the catheter, which eventually results in the complete blockage of the catheter [7,16,17]. Previous attempts to prevent CAUTIs include improving sterile techniques to inhibit the access of microbes into the urinary tract and limiting microbial accumulation on the catheter surface by intermittent catheterization. However, clinical evidence shows that these efforts do not lead to a noticeable reduction in CAUTIs [15]. Therefore, developing novel urinary catheters with antibiofilm and antiencrustation properties remains the most direct and promising strategy for this significant clinical problem.

**Table 1 marinedrugs-19-00255-t001:** Common pathogens causing CAUTI.

**Short-Term Catheterization**	**Type**	**References**
*Escherichia coli*	GN bacterium	[18]
*Serratia* spp.	GN bacterium	[19]
*Staphylococcus epidermidis*	GP bacterium	[20]
*Enterococcus* spp.	GP bacterium	[21]
*Bacillus subtilis*	GP bacterium	[3]
**Long-Term Catheterization**	**Type**	**References**
*Providencia aeruginosa*	GN bacterium	[1]
*Proteus mirabilis*	GN bacterium	[7]
*Providencia stuartii*	GN bacterium	[22]
*Morganella morganii*	GN bacterium	[6]
*Klebsiella pneumoniae*	GN bacterium	[23]
*Staphylococcus aureus*	GP bacterium	[24]
*Candida* spp.	Fungus	[25]

GN: Gram-negative; GP: Gram-positive.

## 2. Current Anti-Infection Strategies against CAUTIs and Challenges

Current commercial urinary catheters can be generally classified as standard and antimicrobial catheters (Figure 2). Owing to their superior malleability, several materials used for making standard catheters include polyvinyl chloride (PVC), polyurethane (PU), silicone and latex [20]. Of these, silicone has emerged as the material of choice for urinary catheters due to distinct advantages, including excellent biocompatibility, no allergic reactions, superior chemical and thermal stability and good mechanical strength [2,26]. Morris et al. [27] compared the antiencrustation performance of 18 types of catheter and found that all-silicone urinary catheters took the longest time to encrust and block. The main reason for this lies in that the all-silicone catheter has a wider lumen, allowing a faster urine flow, which prevents the accumulation of crystalline deposits. However, recent studies demonstrated that there was no significant difference between the development of infections or bacterial adherence on silicone catheters as compared to other types of catheters [20,26,28,29]. 

Given that bacterial adhesion is the critical step in the progression of biofilm formation, numerous attempts have been made to endow the catheters with antiadhesive or antimicrobial properties, or both. Antiadhesive coatings are designed to prevent microbial adhesion through mechanisms of steric repulsion, electrostatic repulsion or low surface energy instead of killing the microbes [15]. To date, hydrogel- and polytetrafluoroethylene (PTFE)-coated catheters are commercially available, but clinical studies demonstrate that their efficacies against CAUTI are insignificant when compared with standard catheters [2,4,26,30]. Despite their lubricating features that may help improve patient comfort, hydrogel- or PTFE-coated catheters are only suitable for short- or medium-term catheterization (<28 days). Antimicrobial coatings are characterized by bactericidal or bacteriostatic activities that protect the catheters from microbial adhesion and migration. Recent reviews of commercial antimicrobial catheters identified two main strategies used: silver-based coatings and antibiotic impregnation [1,7,31]. However, all these catheters have only been reported to yield positive results for short-term application [7,20,32]. 

Silver is among the few FDA-approved antimicrobial materials for urinary catheter coatings, and its antimicrobial activity is associated with the release of silver ions. To date, silver has been applied in catheter coatings in the forms of bulk silver, silver alloy (silver/gold or palladium) and silver-hydrogels. Silver is very prone to oxidation in aqueous conditions, and the release of silver ions from these coatings often undergoes an initial burst-release phase followed by a slow-release phase. In clinical trials, the long-term antimicrobial efficacy of these silver-based coatings has proven limited [1,28,33]. In this scenario, attempts have been made to introduce silver nanoparticles into catheter coatings to attain enhanced antimicrobial efficacy. However, concerns have also been raised about the potential toxicity towards patients due to the fast and excessive release of silver ions [33,34]. In comparison, despite studies suggesting that the overuse of antibiotics may result in the development of antibiotic resistance, certain types of antibiotic-impregnated catheters have been proven to be more effective than silver-based antimicrobial catheters in preventing CAUTIs [3,20,35,36,37]. For example, nitrofurazone-impregnated catheters are commercially available, and studies comparing silver-alloy coated catheters, silver-hydrogel coated catheters, and nitrofurazone-impregnated catheters have found that nitrofurazone could effectively reduce the risk of symptomatic CAUTI and bacteriuria in short-term catheterization by impairing bacterial adherence and planktonic growth, while silver-based catheters have only demonstrated minimal effects [28,38]. Pickard et al. [33] compared the ability of silver-alloy-coated catheters and nitrofural-impregnated catheters for the reduction in incidence of symptomatic CAUTIs in adults requiring short-term catheterization via a clinical model, and the results demonstrated that nitrofural-impregnated catheters were more effective than the silver-alloy-coated catheters. In addition, impregnating antibiotics into catheters provides a cost-effective strategy to manufacture antimicrobial catheters, as this can be achieved by simply submerging swollen catheters in antibiotic-containing solutions for antibiotic encapsulation [31]. However, there are still certain concerns about using nitrofurazone in urinary catheters, such as patient discomfort [38] and the potential risks of developing tumours [39] and resistance in bacteria. This has hindered research in this field, and there is a growing demand for developing new antibiotics instead. 

Apart from silver and antibiotics, recent research has also focused on a variety of antifouling materials (e.g., poly(ethylene glycol) (PEG) and polyzwitterions) [2,40] and biocidal materials (e.g., nitric oxide, antimicrobial peptides, enzymes and bacteriophages) [41,42,43,44] for urinary catheter coatings and has reported with varying levels of success, which offers potential for complete protection against CAUTIs. However, the promising performance of anti-infection coatings under laboratory conditions has not been translated into clinical success to date. Therefore, exploring novel anti-infection materials for urinary catheters will remain a current and broad interest in the upcoming decade.

## 3. Marine Microbiota as a Source of Novel Anti-Infection Materials

For short-term urinary catheters, the use of antibiotics has proven to be a cost-efficient strategy to prevent CAUTIs in clinical trials, but a major concern is the development of antibiotic resistance, particularly the increasing emergence of new forms of multidrug resistance among uropathogens that can render these antibiotics useless after repeated applications. Therefore, the identification of new resources for novel antibiotics with new modes of action has become the research focus over the past two decades. Currently, over 75% of the antibiotics currently available on the market are derived from terrestrial organisms, and the discovery of new antibiotics has declined considerably (only two new classes of antibiotics have been commercialized since 1962) due to difficulties in identifying novel and effective compounds [45,46,47]. 

Oceans cover 71% of the Earth’s surface, but less than 5% have been explored to date, presenting an unexplored area of opportunity [48,49]. In recent decades, significant progress in the clinical development of marine-derived drugs has been achieved, and the discovery of novel antimicrobial substances from marine organisms has indicated their potential to combat existing medical device-related infections [50,51,52,53,54]. However, there are concerns that the overexploration of marine organisms may affect the balance between ecological constraints and economic activity. In this scenario, marine microbiota are emerging as a viable source of bioactive materials [55]. Diverse marine habitats provide unique conditions for marine microorganisms to develop into complex and diverse assemblages, and the isolation and extraction of bioactive secondary metabolites from such organisms are relevant to the discovery of novel anti-infection agents [56,57,58,59]. Therefore, microbes isolated from previously unexplored marine habitats may lead to the discovery of novel structures with potent antibiotic activity, and marine microbial-derived antibiotics are believed to be a promising alternative to overcome existing problems [60,61,62].

For long-term catheters, microbes are more prone to colonize and build biofilms on the surfaces with the attenuation of antimicrobial efficacy. Catheters with only biocidal activity are considered insufficient to eradicate CATUIs, as recent studies demonstrated that a ’foundation layer’ composed of both dead and live bacteria can form on the catheter surface in long-term catheterization, which protects bacteria from contact with underlying biocidal substances [4,63]. To solve this problem, more research has focused on endowing catheter surfaces with both biocidal and antiadhesion properties [4,9,15,64]. Biosurfactants (BSs) are amphipathic secondary metabolites produced by microorganisms and have become a promising anti-infection material for medical applications [65,66,67]. Recent studies of the anti-infection functions of biosurfactants have identified three main routes, including killing microbes or inhibiting microbial growth, resisting microbial adhesion and disrupting biofilm formation [68,69,70]. For example, several BSs isolated from terrestrial-derived microbes, such as surfactin [71] and iturin [72], display potent antimicrobial activities, which make them relevant molecules in combating infections. Van Hoogmoed et al. [73] reported that biosurfactants released by *Streptococcus thermophilus* significantly inhibited *Candida* spp. adhesion to silicone rubber, which indicates their potential for use as a defence against colonizing strains on urinary catheters. These biosurfactants are mostly obtained from terrestrial-derived microorganisms, while biosurfactants produced by marine microorganisms have been less explored. Over the last decade, marine microbial-derived biosurfactants have attracted increasing interest, as marine microbes may exhibit unique metabolic and physiological capabilities producing novel metabolites with potent biological activities. To date, a considerable number of marine microbial-derived biosurfactants with antimicrobial, antiadhesive and antibiofilm activities have been obtained, and several have been proven to be effective against a broad spectrum of uropathogens, including Gram-positive and Gram-negative bacteria, as well as the yeast *Candida albicans*, which could potentially offer a new solution to the problems associated with long-term catheterization.

In this review, we summarize the diversity of marine microbial-derived antibiotics and biosurfactants discovered over the past 10 years and discuss their potential for use in combatting CAUTIs in short-term and long-term urinary catheters. 

## 4. Marine Microbial-Derived Antibiotics with Broad-Spectrum Antimicrobial Activity 

Antibiotics are low-molecular-weight compounds produced by microorganisms that kill or inhibit the growth of other microorganisms at low concentrations [74]. The current understanding of the biological activity of antibiotics is mainly cantered on their primary cellular targets. The mechanisms of antibiotic action can be classified into 5 categories: (1) inhibition of cell wall synthesis (the most common mechanism), (2) inhibition of protein synthesis, (3) alteration of cell membranes, (4) inhibition of nucleic acid synthesis and (5) antimetabolite activity [75,76]. However, not all antibiotics have broad-spectrum antimicrobial activity, and their efficiency decays along with catheterization due to a limited shelf-life. Some microbes may be inherently resistant to certain antibiotics, while they may also mutate and/or exchange genetic material with other microbes, leading to the development of multidrug resistance [77,78]. Furthermore, some antibiotics may exhibit toxicity at or close to their therapeutic dose [47]. Considering the diversity of uropathogen species, the ideal antibiotic for urinary catheters should exhibit broad-spectrum antimicrobial activity and avoid developing resistance without inducing cytotoxicity in patients. 

Generally, antibiotic compounds based on their structures and/or biosynthetic origins can be classified as alkaloids [79], quinones [80], phenols [81], polyketides [82], terpenes [83], polyketides [84,85] and peptides [86]. Table 2 lists the recent discoveries of marine microbial-derived compounds with broad-spectrum antimicrobial activity. Although very few of them have been developed into clinical trial phases, several have demonstrated to be effective against a broad spectrum of uropathogens, including Gram-positive and Gram-negative bacteria, fungi, as well as methicillin-resistant *Staphylococcus aureus* (MRSA). Therefore, they could be an alternative to conventional antibiotics for short-term catheters against CAUTIs caused by those pathogens.

Over the last two decades, the most studied group of microbes producing antimicrobial substances is the Firmicutes phylum (particularly the genus *Bacillus*) [87]. Tareq et al. [88,89,90,91] reported the discovery of four types of bioactive molecules with broad-spectrum antimicrobial activity against both Gram-positive and Gram-negative bacteria and fungi. Gageostatins A–C (Figure 3a) and gageopeptides A–D (Figure 3b) are two types of non-cytotoxic linear lipopeptides isolated from a marine *Bacillus subtilis* 109GGC020. The authors proposed a biosynthetic pathway based on nonribosomal peptide synthetases (NRPS) for the production of gageotetrins. Comparative studies showed that gageostatins A–C were more active against fungi than bacteria with minimum inhibitory concentration (MIC) values of 0.02–0.04 μM. Gageopeptides A–D exhibited potent antifungal and moderately broad antibacterial activity, while not showing cytotoxicity to human myeloid leukaemia K-562 and mouse leukemic macrophage RAW 264.7 cell lines. Gageomacrolactins A–C (Figure 3c) are three macrolactin derivatives obtained from the secondary metabolites of marine *Bacillus subtilis* 109GGC020 and displayed strong broad-spectrum activity against Gram-negative and Gram-positive bacteria and fungi. By comparing the structure-function of the gageomacrolactins A–C with that of known macrolactins 4–7, the authors demonstrated that antibacterial activity was not affected by the position of the epoxide group but highly dependent on the hydroxyl group at C-15 of the macrolactone ring. Ieodoglucomides 1 and 2 (Figure 3d) are two unique glycolipopeptides produced by a marine *Bacillus licheniformis* 09IDYM23 which act as moderate antimicrobial molecules. Podilapu et al. [92] reported the successful synthesis of ieodoglucomides A and B through a high-yielding route using per-O-TMS glucosyl iodide, making them promising candidates for industrial application.

Apart from *Bacillus* spp., Uzair et al. [93] reported the isolation of 4-[(Z)-2 phenyl ethenyl] benzoic acid (kocumarin) from the marine *Kocuria marina* CMG S2, which exhibited pronounced and rapid growth inhibition against fungi and pathogenic bacteria, including methicillin-resistant *Staphylococcus aureus* (MRSA). Although its antimicrobial mechanism has not been clearly elucidated, research on the functional groups involved in the molecular mechanism would reveal further insights into this atypical class of antibiotics. Schumacher et al. [84] described the first antimicrobial polyketide (bonactin) isolated from the liquid culture of a *Streptomyces* sp. BD21-2, a compound displaying antifungal and broad-spectrum antibacterial activity against microbes, including *Bacillus megaterium*, *Micrococcus luteus*, *Klebsiella pneumoniae*, *Staphylococcus aureus*, *Alcaligenes faecalis*, *Escherichia coli* and *Saccharomyces cerevisiae*. However, such marine microbial-derived compounds with activity against both bacteria and fungi are still very rare to date. 

Over the past 10 years, numerous molecules possessing broad-spectrum antibacterial or antifungal activities have been isolated from marine microorganisms. Glycosylated macrolactins A1 and B1 (Figure 3e) were isolated from a marine *Streptomyces* sp. KJ371985, which inhibited *Bacillus subtilis*, *Escherichia coli*, *Pseudomonas aeruginosa* and *Staphylococcus aureus* with MICs of 0.03–0.22 μM. These compounds inhibit the peptidyl transferase activity and binding of the acceptor substrate to bacterial ribosomes. The higher solubility in the polar solvents of sugar-containing macrolactins could be an advantage in clinical applications [94]. Bacteriocins are natural peptides synthesized by bacteria for the purpose of killing/inhibiting other bacterial strains whilst not harming the producing bacteria through specific immunity proteins [95]. Elayaraja et al. [96] purified a bacteriocin from marine *Lactobacillus murinus* AU06, which exhibited a broad inhibitory spectrum against both Gram-positive and -negative bacteria. Marinocine is a broad-spectrum antibacterial protein synthesized by the melanogenic marine bacterium *Marinomonas mediterranea*, which generates hydrogen peroxide that kills bacteria [97]. However, recent studies demonstrated that the molecular basis of the antibacterial activity was L-lysine dependent, and activity was inhibited under anaerobic conditions. Mollemycin A (Figure 3f) is an antibacterial glyco-hexadepsipeptide-polyketide isolated from an Australian marine-derived *Streptomyces* sp. (CMB-M0244), which exhibited exceptionally potent and selective growth inhibitory activity against Gram-positive and Gram-negative bacteria (IC50 10–50 nM) but did not show any antifungal activity against *Candida albicans*. The cytotoxicity test also showed that mollemycin A was proportionately less cytotoxic toward human neonatal foreskin fibroblast cells [98]. Thiomarinols A–G (Figure 3g) were discovered as a class of polyketide antibiotics from marine *Alteromonas rava* SANK 73390, which displayed broad-spectrum activity against Gram-positive and Gram-negative bacterial species [99]. These thiomarinols act through inhibiting bacterial isoleucyl-transfer RNA synthetase and have pronounced activity against MRSA, with MICs ≤ 0.01 μg/mL [60]. Similar antibacterial compounds were also obtained from actinobacteria, such as *Pseudonocardia carboxydivorans* M-227 [100] and *Streptomyces* sp. JRG-04 [101]. Fungi (primarily *Candida species*) account for 20–30% of CAUTI cases, and *Candida albicans* is the most prevalent pathogen found in CAUTI biofilms [1,37]. Antifungal (particularly anti-*Candida*) compounds derived from marine microbes have been reported for metabolites produced by *Streptomyces* sp. ZZ338 [102], *Streptomyces* sp. SNM55 [103], *Bacillus subtilis* KC433737 [104], *Janthinobacterium* spp. ZZ145 and ZZ148 [105], *Trichoderma* sp. MF106 [106] and *Stagonosporopsis cucurbitacearumstrain* G019 [107]. These microbial strains, respectively produced actinomycins D, V and X_0β_ (Figure 4a) (MIC, 9.83–9.96 μM); mohangamides A and B (Figure 4b) (IC50, 4.4 and 20.5 μM); 5-hydroxymethyl-2-furaldehyde (5HM2F) (Figure 4c) (MBIC, 400 μg/mL); janthinopolyenemycin A and B (Figure 4d) (MIC, 15.6 μg/ML, MBC, 31.25 μg/mL); trichodin A (Figure 4e) (IC50, 25.38 ± 0.41 μM); and didymellamide A (Figure 4f) (MIC, 3.1 μg/mL).

**Figure 3 marinedrugs-19-00255-f003:**
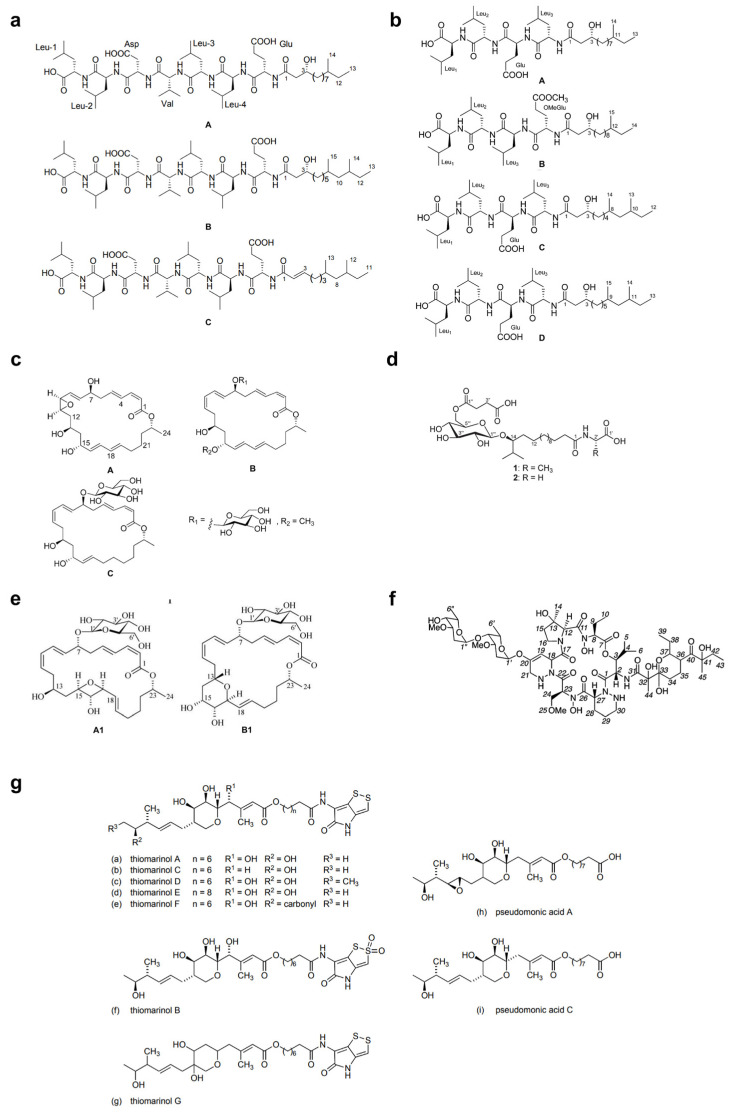
Structures of (**a**) gageosatins A–C [90], (**b**) gageopeptides A–D [91], (**c**) gageomacrolactins A–C [89], (**d**) ieodoglucomides 1 and 2 [88], (**e**) glycosylated macrolactins A1 and B1 [94], (**f**) mollemycin A [98], and (**g**) thiomarinols A–G [99].

**Table 2 marinedrugs-19-00255-t002:** Marine microbial-derived compounds with broad-spectrum antimicrobial activity.

Compound	Molecular Class	Source	Target	Reference
Gageotetrins A–C	Peptide	*Bacillus subtillis* 109GGC020	B/F	[90]
Gageopeptides A–D	Peptide	*Bacillus subtillis* 109GGC020	B/F	[91]
Ieodoglucomide 1, 2	Peptide	*Bacillus licheniformis* 09IDYM23	B/F	[88]
Bacteriocin	Peptide	*Lactobacillus murinus* AU06	B	[96]
Actinomycins D, V, X_0β_	Peptide	*Streptomyces* sp. ZZ338	F	[102]
Mohangamides A, B	Peptide	*Streptomyces* sp. SNM55	F	[103]
Gageomacrolactins A–C	Macrolide	*Bacillus subtillis* 109GGC020	B/F	[89]
Glycosylated macrolactins A1, B1	Macrolide	*Streptomyces* sp. (KJ371985)	B	[94]
Bonactin	Acyclic ester	*Streptomyces* sp. BD21-2	B/F	[86]
Butenolide	Lactone	*Streptomyces* sp.	B	[108]
Mollemycin A	Peptide-polyketide	*Streptomyces* sp. CMB-M0244	B	[98]
Thiomarinols A-G	Polyketide	*Alteromonas rava* SANK 73390	B	[99]
Branimycin B, C	Polyketide	*Pseudonocardia carboxydivorans* M-227	B	[100]
UN	Polyketide	*Streptomyces* sp. JRG-04	B	[101]
Janthinopolyenemycin A, B	Polyketide	*Janthinobacterium* spp. ZZ145 and ZZ148	F	[105]
Kocumarin	Benzoic acid	*Kocuria marina* CMG S2	B/F	[93]
Marinocine	Protein	*Marinomonas mediterranea* MMB-1	B	[109]
5HM2F	Furan	*Bacillus subtilis* KC433737	F	[104]
Trichodin A	Pyridone	*Trichoderma* sp. MF106	F	[106]

B: Gram-positive and Gram-negative bacteria; F: fungi; UN: unnamed material.

## 5. Marine Microbial-Derived Biosurfactants: New Agents against CAUTIs

Biosurfactants (BSs) are amphipathic secondary metabolites produced by microorganisms that consist of both hydrophilic and hydrophobic moieties [45]. Compared to conventional chemical surfactants, these biomolecules have many advantages, such as low toxicity; high biodegradability; biocompatibility; low critical micelle concentrations (CMC); an ability to function over wide ranges of pH, temperature and salinity; as well as greater selectivity, and can be produced from renewable, cheaper substrates [110,111]. Such characteristics allow BSs to play a key role in multidisciplinary applications in industrial and environmental fields and make them a green alternative to their chemical counterparts. Furthermore, some BSs have been reported for their specific bioactivities, which play an essential role in the survival of microbial producers against other competing microbes [69,112]. In nature, BSs can be secreted extracellularly or remain attached to cell surfaces, reducing surface tension at the interface, thereby reducing surface contamination and aiding microbial motility in potentially hostile environments [113]. BSs may also have a range of therapeutic and biomedical benefits and could be used instead of conventional antibiotics to combat infections [114]. Current research regarding BSs has mostly focused on microbes from terrestrial sources (particularly soil-isolated microbes such as species of *Bacillus*, *Pseudomonas* and yeasts) [45,112], while practical applications of BSs in healthcare are still limited [115]. For this reason, marine habitats have again emerged as a potential source for isolating new BSs, and the discovery of new BS-producing microorganisms is attracting increasing interest.

BSs according to their molecular weight can be grouped into two categories: (a) high-molecular-mass (HMW) molecules (also called bioemulsifiers), such as polysaccharides, proteins, lipopolysaccharides, lipoproteins and lipoheteropolysaccharides, and (b) low-molecular-mass (LWM) molecules (generally 500 to 1500 Da), which can be further subdivided into glycolipids, lipopeptides, phospholipids, polymeric compounds and neutral lipids based on their chemical composition and microbial origin [45,65,116]. Recent research has mainly focused on the LWM BSs, and several have been reported to display antimicrobial, antiadhesive and antibiofilm activities against a broad spectrum of uropathogens (including multidrug-resistant pathogens), indicating their use as promising anti-infection materials for urinary catheters (Table 3). 

### 5.1. Antimicrobial Activity

Biosurfactants have been reported to exhibit antimicrobial activity via different mechanisms of action, which primarily destroy the cell wall or plasma membrane by disrupting their integrity and permeability [65,68,117,118]. More specifically, their amphiphilic characteristics and affinity for lipid bilayers allow BSs to interact with cell membranes, leading to cell lysis and metabolite leakage, which ultimately results in cell death [116]. To date, various marine microbes have been reported to produce antimicrobial BSs, of which glycolipids and lipopeptides are the two primary isolated families that display broad-spectrum antimicrobial activity. 

Glycolipids are a class of carbohydrate molecules made of mono-, di-, tri- and tetra-saccharides in combination with long-chain aliphatic acids or hydroxyaliphatic acids [119], of which trehalolipids, rhamnolipids and sophorolipids are of the most interest [120]. Reported marine microbial-derived glycolipid BSs with broad-spectrum antimicrobial activity against both bacteria and fungi are listed in Table 3. A glycolipid BS isolated from a marine *Staphylococcus saprophyticus* SBPS 15 exhibited dual functions: (1) excellent antibacterial and antifungal activities against a broad spectrum of clinical human pathogens, including Gram-positive bacteria (*Bacillus subtilis* and *Staphylococcus aureus*), Gram-negative bacteria (*Escherichia coli*, *Pseudomonas aeruginosa*, *Klebsiella pneumoniae* and *Salmonella paratyphi*) and fungi (*Aspergillus niger*, *Candida albicans* and *Cryptococcus neoformans*), and (2) potent surface tension-reducing activity (32 mN/m). This BS also showed superior stability over a broad range of pH (3–9) and temperature (up to 80 °C) [121]. Another glycolipid BS purified from extracts of a tropical marine strain of *Serratia marcescens* demonstrated similar dual functions, inhibiting the growth of *Candida albicans* and *Pseudomonas aeruginosa* with MIC values of >25.0 μg/mL and preventing adhesion by up to 99% [122]. The glycolipid BS also displayed biofilm disrupting activities against the test strains, which were believed to result from synergistic antimicrobial and surfactant activity. Glycolipid BSs were also isolated from the marine actinobacteria *Brevibacterium casei* MSA19, *Streptomyces* sp. MAB36 and *Brachybacterium paraconglomeratum* MSA21. The MSA19 glycolipid was bacteriostatic and could inhibit microbial attachment and disrupt both fungal and bacterial biofilms in both individual strains and mixed cultures at 30 µg/mL [123]. The MAB36 glycolipid also possessed strong antimicrobial activity against pathogenic bacteria and fungi and demonstrated excellent stability over a wide range of pH, temperature and ion concentrations [124]. The BS produced by the sponge-associated *Brachybacterium paraconglomeratum* MSA21 displayed broad antibiotic activity and it was suggested that the discovery of polyketide synthase (pks II) genes might pave the way for making new BSs by exploiting the microbial genes and enzymes, making the green production of BSs possible in the future [125]. Another glycolipid BS was isolated from a marine halotolerant bacterium (*Buttiauxella* sp. M44) that exhibited significant stability over a broad range of pH (7–8), temperature (20–60 °C) and salinity (0–3%) and potent antimicrobial activity against both bacteria and fungi [126]. Moreover, the combination of a cheap energy and carbon source (e.g., molasses) as well as a response surface method (RSM) could favor production and expand possible uses in the future.

Lipopeptides (LPs) are composed of peptide chains (short linear or cyclic structures) with lipid moieties and are the most widely reported class of biosurfactants with antimicrobial activity. *Bacillus* species are the most studied LP-producing strains that have been reported to produce several antimicrobial lipopeptide biosurfactants (LPBs), such as surfactin, iturin and fengycin, although most of these strains are from terrestrial ecosystems [70,72]. Liu et al. [71] reported the first surfactin (mainly composed of the nC14- and anteiso-C15-surfactin) isolated from marine *Bacillus velezensis* H3, which exhibited antimicrobial activity against a broad range of pathogens, including *Staphyloccocus aureus*, *Klebsiella peneumoniae*, *Pseudomonas aeruginosa* and *Candida albicans*. The surfactin demonstrated a lower inhibitory effect than the antibiotic polymixin B but greater antifungal activity against *Candida albicans* than the antibiotic vancomycin. Similar antibacterial and antifungal activities were also found for the BSs produced by *Halobacterium salinarum* [127]. Another surfactin produced by the marine actinobacterium *Nocardiopsis alba* MSA10 was shown to effectively inhibit the growth of *Enterococcus faecalis*, *Klebsiella pneumoniae*, *Micrococcus luteus*, *Proteus mirabilis*, *Staphylococcus aureus*, *Staphylococcus epidermidis* and *Candida albicans*, however, without effects on *Escherichia coli* and *Pseudomonas aeruginosa* [128]. Apart from surfactin, Balan et al. [129] first reported the discovery of aneurinifactin from the marine *Aneurinibacillus aneurinilyticus* SBP-11, which showed broad-spectrum antibacterial activity against *Klebsiella pneumoniae*, *Escherichia coli*, *Staphylococcus aureus*, *Pseudomonas aeruginosa*, *Bacillus subtilis* and *Vibrio cholerae*. The antibacterial mechanism was proposed to be derived from the anchoring of the BS on the bacterial cell membrane that disrupted its integrity, resulting in the production of hydroxyl radicals, which in turn caused lipid peroxidation and pore formation in the membrane [129]. However, antifungal activity was not investigated. It was also reported that a LPB (pontifactin) produced by marine *Pontibacter korlensis* SBK-47 displayed high antibacterial activity against *Streptococcus mutans*, *Micrococcus luteus*, *Salmonella typhi* and *Klebsiella oxytoca* and moderate activity against *Klebsiella pneumonia* and *Vibrio cholerae*. The molecule also showed antiadhesion potential against *Bacillus subtilis*, *Staphylococcus aureus*, *Salmonella typhi* and *Vibrio cholerae* [130]. Lawrance et al. [131] discovered an antibacterial LPB from the marine sponge-associated bacterium *Bacillus licheniformis* NIOT-AMKV06, which displayed significant bacteriostatic activity against a range of human pathogens, including *Enterococcus faecalis*, *Klebsiella pneumoniae*, *Micrococcus luteus*, *Proteus mirabilis*, *Salmonella typhi*, *Shigella flexneri*, *Staphylococcus aureus* and *Vibrio cholerae*. The production of these biosurfactants in heterologous host strains was achieved by cloning three gene clusters (sfp, sfpO and srfA) in *Escherichia coli*, with production being increased three-fold over the original strain. In addition, a lipopeptide BS produced by the marine *Bacillus circulans* is the only one found to be effective against multidrug-resistant (MDR) clinical strains while not having any haemolytic activity, indicating its potential to combat infections caused by such pathogens [132].

### 5.2. Antiadhesive and Antibiofilm Activities

Biosurfactants have also been reported to inhibit microbial adhesion and biofilm formation without exerting antimicrobial activity [69,133,134]. Despite the precise mechanisms of such activity not being fully understood, biosurfactants may affect interactions between microbes and surfaces through several modes of action: (1) modification of the physico-chemical properties (e.g., surface charge, hydrophobicity and surface energy) of the surface, which reduces microbial adhesion [111]; (2) suppression of the expression of biofilm-related genes, which inhibits microbial adhesion [135]; (3) promotion of the solubilization of biofilms encouraging bacterial detachment [136]; and (4) the interference of quorum sensing leading to decreased biofilm formation [70]. Numerous studies have shown that the prior adhesion of BSs to catheter surfaces (surface conditioning) reduced microbial adhesion and colonization [137,138,139].

In general, bacterial adhesion to a surface is regulated by diverse factors (e.g., growth medium, substrate and cell surface). The most frequently cited theory is the two-step adhesion model in which the adhesion process is divided into two distinct phases: reversible adhesion and irreversible adhesion [11,12,140]. Reversible adhesion describes a dynamic process in which bacteria can easily attach to or detach from a surface due to a combination of surface–cell interactions including van der Waals, electrostatic double-layer, Lewis acid-base, Brownian motion and hydrophobic interactions [64,141]. Walencka et al. [142] reported that BSs can affect cell-to-surface interactions by altering surface tension and charge to overcome the initial electrostatic repulsion barrier. For instance, Meylheuc et al. [143] reported that a stainless-steel surface coated with the BS produced by *Pseudomonas fluorescens* significantly reduced microbial adhesion. The presence of BS in the conditioned surface remarkably reduced the surface energy and enhanced surface hydrophilicity, leading to a decrease in attraction due to a reduction in the van der Waals forces and an increase in electron–donor/electron–acceptor characteristics. Jemil et al. [144] also reported that electrostatic repulsion between negatively charged surfaces (coated with anionic lipopeptides) and the negatively charged microbial surface could aid in inhibiting microbial adhesion. However, following reversible adhesion, microbial attachment gradually becomes stronger with time through a range of interfacial rearrangements (e.g., removal of interfacial water, protein conformational changes and an increase in hydrophobic interactions) and the production of adhesins, eventually form biofilms [145]. Despite numerous studies highlighting the difficulties in eliminating biofilms, BSs have been shown to disrupt or remove biofilms by penetrating and absorbing at the interface between the solid substrate and the biofilm, thereby reducing interfacial tension [134,146]. 

Antiadhesive and antibiofilm activities have also been reported for marine microbial-derived BSs. Hamza et al. [134] reported a non-toxic glycolipid BS (SLSZ2) derived from a marine epizootic bacterium *Staphylococcus lentus* SZ2 that effectively prevented the adhesion of *Vibrio harveyi* and *Pseudomonas aeruginosa* without showing bactericidal activity. In addition, this BS efficiently inhibited biofilm formation and disrupted the pre-formed biofilms of both strains by ~80%. Another glycolipid BS produced from a marine *Symphylia* sp. also exhibited antiadhesive activity against a range of pathogens (*Candida albicans*, *Pseudomonas aeruginosa* and *Bacillus pumilus*) and could disrupt pre-formed biofilms of these cultures in a concentration-dependent manner [122]. Similarly, a glycolipid BS isolated from the marine actinobacterium *Brevibacterium casei* MSA19 disrupted biofilm formation in *Escherichia coli*, *Pseudomonas aeruginosa* and *Vibrio* spp. under dynamic conditions. Moreover, biofilm disruption activity was consistent against mixed and individual biofilm bacteria at ~30 µg/mL. The lipopeptide BS produced by marine *Bacillus circulans* exhibited promising antiadhesive activity against several potential pathogenic strains, and the BS-coated surface effectively reduced microbial adhesion by up to 89% at a concentration of 0.1 g/L. Moreover, the pre-formed biofilms were removed with efficiencies between 59 and 94% for all the strains tested, demonstrating its potential in biomedical applications [147]. Song et al. [148] reported a lipopeptide BS produced by marine *Bacillus amyloliquefaciens* anti-CA that inhibited biofilm formation and dispersed pre-formed biofilms of *Pseudomonas aeruginosa* PAO1 and *Bacillus cereus*. The obtained data indicated that the BS could suppress the expression of the PslC gene which is associated with exopolysaccharide production in *Pseudomonas aeruginosa* PAO1. Pradhan et al. [149] also reported a lipopeptide BS produced by a marine *Bacillus tequilensis* CH which effectively inhibited pathogenic biofilms (*Escherichia coli* and *Streptococcus mutans*) on both hydrophilic and hydrophobic surfaces at a concentration of 50 μg/mL. 

**Table 3 marinedrugs-19-00255-t003:** Marine microbial-derived BSs with antimicrobial, antiadhesive or antibiofilm activities.

Source	Type	Activity	Reference
*Staphylococcus saprophyticus* SBPS 15	Glycolipid	Antibacterial activity against *Klebsiella Pneumoniae*, *Escherichia coli*, *Pseudomonas aeruginosa*, *Bacillus subtilis*, *Salmonella paratyphi* and *Staphylococcus aureus* Antifungal activity against *Aspergillus niger*, *Candida albicans* and *Cryptococcus neoformans*	[121]
*Serratia marcescens*	Glycolipid	Antibacterial activity against *Pseudomonas aeruginosa* and *Bacillus pumilus* Antifungal activity against *Candida albicans* Antiadhesive activity against *Pseudomonas aeruginosa*, *Bacillus pumilus* and *Candida albicans*	[122]
*Brevibacterium casei* MSA19	Glycolipid	Antibacterial activity against *Escherichia coli*, Klebsiella pneumoniae, *Proteus mirabilis*, *Pseudomonas aeruginosa*, *Vibrio parahaemolyticus* and *Vibrio vulnificus* Antibiofilm activity against mixed and individual cultures of *Escherichia coli*, *Pseudomonas aeruginosa* and *Vibrio* spp.	[123]
*Streptomyces* sp. MAB36	Glycolipid	Antibacterial activity against *Bacillus cereus*, *Enterococcus faecalis*, *Proteus mirabilis*, *Pseudomonas aeruginosa*, *Staphylococcus aureus*, *Staphylococcus epidermidis*, *Shigella dysenteriae* and *Shigella boydii* Antifungal activity against *Candida albicans*	[124]
*Brachybacterium paraconglomeratum* MSA21	Glycolipid	Antibacterial activity against *Bacillus subtilis*, *Escherichia coli*, *Enterococcus faecalis*, *Klebsiella pneumoniae*, *Micrococcus luteus*, *Pseudomonas aeruginosa*, *Proteus mirabilis*, *Streptococcus* sp., *Staphylococcus aureus* and *Staphylococcus epidermidis* Antifungal activity against *Candida albicans*	[125]
*Buttiauxella* sp. M44	Glycolipid	Antibacterial activity against *Escherichia coli*, *Salmonella enterica*, *Bacillus cereus*, *Bacillus subtilis* and *Staphylococcus aureus* Antifungal activity against *Candida albicans* and *Aspergillus niger*	[126]
*Staphylococcus lentus* SZ2	Glycolipid	Antiadhesive activity against *Vibrio harveyi* and *Pseudomonas aeruginosa* Antibiofilm activity against *Vibrio harveyi* and *Pseudomonas aeruginosa*	[134]
*Bacillus velezensis* H3	Lipopeptide	Antibacterial activity against *Staphyloccocus aureus*, *Mycobacterium*, *Klebsiella peneumoniae* and *Pseudomonas aeruginosa* Antifungal activity against *Candida albicans*	[71]
*Halobacterium salinarum*	Lipopeptide	Antibacterial activity against *Escherichia coli*, *Bacillus* sps., *Pseudomonas* sp., *Streptococcus* sp. And *Staphylococcus aureus* Antifungal activity against *Aspergillus niger* and *Candida albicans*	[127]
*Nocardiopsis alba* MSA10	Lipopeptide	Antibacterial activity against *Enterococcus faecalis*, *Klebsiella pneumoniae*, *Micrococcus luteus*, *Proteus mirabilis*, *Staphylococcus aureus* and *Staphylococcus epidermidis* Antifungal activity against *Candida albicans*	[128]
*Aneurinibacillus aneurinilyticus* SBP-11	Lipopeptide	Antibacterial activity against *Klebsiella pneumoniae*, *Escherichia coli*, *Staphylococcus aureus*, *Pseudomonas aeruginosa*, *Bacillus subtilis* and *Vibrio cholerae*	[129]
*Bacillus licheniformis* NIOT-AMKV06	Lipopeptide	Antibacterial activity against *Enterococcus faecalis*, *Klebsiella pneumoniae*, *Micrococcus luteus*, *Proteus mirabilis*, *Salmonella typhi*, *Shigella flexneri*, *Staphylococcus aureus* and *Vibrio cholera*	[131]
*Pontibacter korlensis* strain SBK-47	Lipopeptide	Antibacterial activity against *Streptococcus mutans*, *Micrococcus luteus*, *Salmonella typhi* and *Klebsiella oxytoca* Antiadhesion potential against *Bacillus subtilis*, *Staphylococcus aureus*, *Salmonella typhi* and *Vibrio cholerae*	[130]
*Bacillus tequilensis* CH	Lipopeptide	Antibiofilm activity against *Escherichia coli* and *Streptococcus mutans*	[149]
*Bacillus Amyloliquefaciens* anti-CA	Lipopeptide	Antibiofilm activity against *Pseudomonas aeruginosa* and *Bacillus cereus*	[148]
*Bacillus circulans*	Lipopeptide	Antiadhesive activity against *Escherichia coli*, *Micrococcus flavus*, *Serratia marcescens*, *Salmonella typhimurium*, *Proteus vulgaris*, *Citrobacter freundii*, *Alcaligenes faecalis*, and *Klebsiella aerogenes*	[147]
*Aspergillus ustus* MSF3	Glycolipoprotein	Antibacterial activity against *Enterococcus faecalis*, *Escherichia coli*, *Klebsiella pneumoniae*, *Micrococcus luteus*, *Pseudomonas aeruginosa*, *Proteus mirabilis*, *Staphylococcus aureus*, *Staphylococcus epidermidis* and *haemolytic Streptococcus* Antifungal activity against *Candida albicans*	[150]
*Streptomyces* sp. B3	Mixture of proteins, carbohydrates and lipids	Antibacterial activity against *Escherichia coli* and *Pseudomonas aeruginosa* Antifungal activity against *Candida albicans*	[151]
*Oceanobacillus iheyensis* BK6	Extracellular polysacchrides	Antibiofilm activity against *Staphylococcus aureus*	[152]

## 6. Opportunities, Challenges and Future Perspectives

Currently, research on anti-infection materials for urinary catheters is still increasing, and there is a growing demand for catheters with stronger antibiofilm and antiencrustation capabilities. Compared to the conservative strategy of optimizing existing products (e.g., widening the internal lumen [1], replacing bulk silver with silver nanoparticles [4], and redesign of micro/nano-scale surface topography [153]), the exploration of novel anti-infection materials from marine resources has clearly become an exciting potential solution to address some current limitations. Owing to the vast diversity of marine microorganisms, a number of new bioactive molecules with antimicrobial, antiadhesive and antibiofilm activities have been discovered that could be potentially combined with urinary catheters to combat CAUTIs. 

The antimicrobial strategy aims to endow urinary catheters with microbicidal or microbiostatic properties to prevent the contact of microbes with the catheter surface or inhibit microbial migration along the catheter, thus preventing biofilm formation and encrustation. For short-term catheters, the use of antibiotics has proven to be the most efficient and cost-effective strategy, whereas the application of conventional antibiotics (e.g., nitrofurazone) has been questioned due to the safety concerns and the presence of antibiotic resistance. Despite attempts having been made to develop new antibiotics to solve this problem, the difficulty in identifying novel and effective compounds has led to a slowdown of research in this field. In recent decades, the isolation of microbes from previously unexplored marine habitats has led to the discovery of a number of novel antimicrobial compounds with new modes of action to combat the current antibiotic resistance threat. Several of these have been proven to possess broad-spectrum antimicrobial activities against a wide range of uropathogens, which could be potentially applied to urinary catheters to eradicate CAUTIs. Technically, these antimicrobial molecules could be directly deposited on the catheter surfaces or applied in coatings. Considering that CAUTIs may be derived from both extraluminal and intraluminal routes, it would be ideal to endow both surfaces with antimicrobial properties, despite extraluminal infections being clinically more common. For example, commercial catheters have been conventionally impregnated with antimicrobials, such as antibiotics, and function via a release model. However, this is only a temporary solution for short-term catheterization, and microbial contamination can resume once the antibiotics are removed. Given that the efficacy of antimicrobial materials is often concentration dependent, a challenge in this type of catheter modification strategy is to achieve and preserve adequate local delivery of the antimicrobials during catheterization in vivo without inducing resistance and patient cytotoxicity. This could be achieved by developing a novel release system with controlled release kinetics instead of directly altering the drug loading. For example, hydrogel coatings, due to their lubricating properties, have been widely applied in urinary catheters. Milo et al. [154] reported a smart infection-responsive hydrogel coating for urinary catheters. The coating is a dual-layered polymeric system consisting of a lower ‘reservoir’ layer of poly(vinyl alcohol) (PVA) hydrogel (containing bacteriophage), capped by an upper layer of the pH-responsive polymer. The upper layer swells when urinary pH is elevated due to *P. mirabilis* infection, exposing the PVA reservoir layer to urine, resulting in the release of bacteriophage from the coating to prevent the encrustation and blockage of urinary catheters. In this way, the release of antimicrobials could be controlled in a smart way to prevent/retard CAUTIs and associated complications. Researchers will need to further conduct thorough in vitro and in vivo leaching and cytotoxicity studies to determine the optimum operational conditions and identify any problematic side effects.

In long-term catheterization, extensive biofilms containing >5 × 10^9^ viable cells/cm^2^ can be found on catheters and can be responsible for the persistence of the infections [155]. Moreover, the biofilms are often composed of multispecies consortia, which have been reported to be more virulent and have a higher tolerance to antibiotics [7]. Therefore, in addition to killing the uropathogens, attempts have also been made to combine catheter surfaces with antiadhesive materials to retard the occurrence of CAUTIs. To date, only PTFE and hydrogel have entered clinical use, while their efficacy in resisting CAUTI and encrustation is still controversial [4,17,33]. Other antiadhesive materials, such as polyzwitterions, have recently emerged as promising candidates, although their instability in long-term catheterization hinders practical application [1]. In this scenario, research on marine microbes has led to the discovery of novel antiadhesive molecules with greater activity and stability. For instance, marine microbial-derived antiadhesive materials, such as biosurfactants and cyanobacterial extracellular polymers [59,156], can inhibit microbial adhesion via multimechanisms. Currently, several biosurfactants have been reported to influence both cell-to-cell and cell-to-surface interactions that can inhibit microbial adhesion as well as disrupt biofilm formation [67,70,112]. Technically, biosurfactants could be doped in coatings and act on microbial cells via a contact mode. These features may aid in preventing biofilm formation and encrustation as well as extending the lifetime of indwelling urinary catheters. Furthermore, they could be employed in combination with other strategies (e.g., use of antimicrobial materials and micro/nano patterning) to achieve synergistic effects. 

In fact, although these marine microbial-derived molecules have shown potential in combating CAUTI-related pathogens and/or biofilms, it should be noted that their clinical effectiveness still needs to be verified in future. Currently, there has been a lack of accomplished research in this area, and this paper provides a prospective proposal for researchers. Currently, the most significant obstacles hindering the development of marine microbial-derived products are supply issues. This is mainly due to difficulties associated with the isolation and growth of producing microorganisms. In addition, for marine microbial-derived biosurfactants, most of these materials are unidentified mixtures of compounds, and the further pharmaceutical development of such mixtures is very challenging. To date, the development of new chemical and physicochemical approaches and tools has led to great advances in the isolation and structure elucidation of novel minor marine secondary metabolites, which could not be isolated/detected in the past [54]. On the other hand, urinary catheters are still considered as low-cost clinical care products, although commercial ‘anti-infective’ catheters are usually priced two to five times higher than standard catheters. Therefore, in addition to the cost of employing new materials, a cost-effective coating technology should also be addressed. 

## 7. Conclusions

Marine microorganisms can produce metabolites of varied chemical structures displaying antimicrobial, antiadhesive and antibiofilm activities. These compounds present enormous potential for the discovery of new agents to overcome the current challenges associated with CAUTI. Owing to technical limitations in the past, bioactive compounds derived from marine microorganisms have not yet progressed into clinical trials. However, recent advances in oceanographic science and metabolome screening have led to a boost in the discovery of new species and metabolic profiles as well as new molecules with potent anti-infection activities against CAUTI-associated pathogens. Research efforts should be applied to the exploration of marine microbes in the search for new bioactive compounds for the development of novel anti-infection agents.

## Figures and Tables

**Figure 1 marinedrugs-19-00255-f001:**
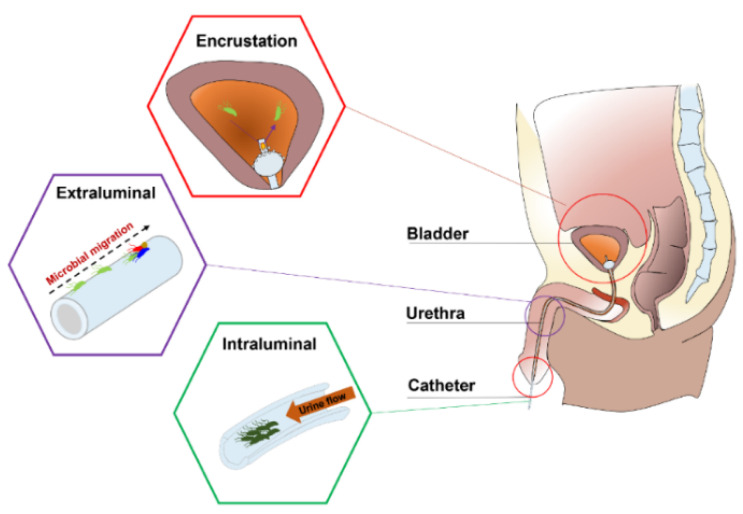
Anatomical cross-section of the renal system in a male showing CAUTIs.

**Figure 2 marinedrugs-19-00255-f002:**
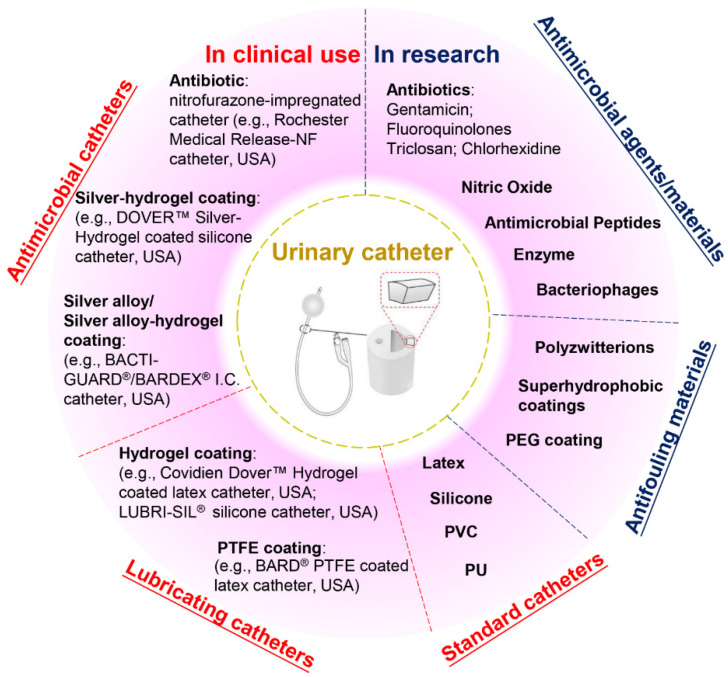
Classification of urinary catheters in clinical use and recently reported antimicrobial and antifouling materials for the prevention of CAUTIs.

**Figure 4 marinedrugs-19-00255-f004:**
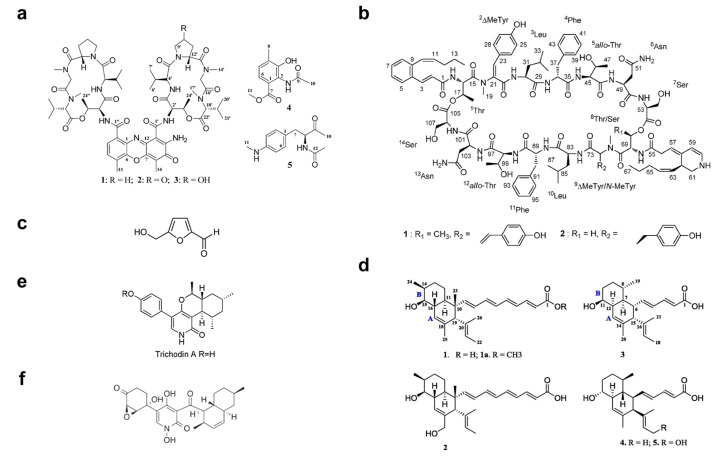
Structures of (**a**) actinomycins D, V and X_0β_ [102], (**b**) mohangamides A and B [103], (**c**) 5-hydroxymethyl-2-furaldehyde (5HM2F) [104], (**d**) janthinopolyenemycin A and B [105], (**e**) trichodin A [106], and (**f**) didymellamide A [107].

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
