# Peer review of "Marine Microbial-Derived Antibiotics and Biosurfactants as Potential New Agents against Catheter-Associated Urinary Tract Infections"

_marinedrugs, 2021, doi:10.3390/md19050255_

Round 1

Reviewer 1 Report

The authors of the manuscript ID: 1188697 entitled "Marine Microbial-Derived Antibiotics and Biosurfactants as Potential New Weapons against Catheter-associated Urinary Tract Infections" presented important clinical and scientific aspects.

In this review summarizes information on the most relevant materials that have been obtained from marine-derived microorganisms over the last decade and discusses their potential as new weapons against catheter-associated urinary tract infections (CAUTIs).
The authors presented the topic of the work in an exhaustive way. They have included summary figures, tables, and structural formulas of chemicals / antibiotics. The manuscript fully deserves publication in the journal Marine Drugs. 

Author Response

Dear Reviewer: 

Many thanks for your comments. We do appreciate your time and support.

All the best

Shuai on behalf of all authors 

Reviewer 2 Report

This is a well written comprehensive text, in perfect English. It will be interesting for a quite narrow public, but for that public, it might be of hight value. 

My only remark concerns some overlap between the first part of the text, and the kind of 'discussion' : ' Opportunities, Challenges and Future Perspectives'. 

Author Response

Point 1:  My only remark concerns some overlap between the first part of the text, and the kind of 'discussion' : ' Opportunities, Challenges and Future Perspectives'. 

Response 1: The overlap in section 6 (Opportunities, Challenges and Further Perspectives) has been deleted. Please see the attachment.

Reviewer 3 Report

This manuscript gives an overview of the marine products potentially usable to address the urethral catheters infections (catheters-associated urinary tract  infections: CAUTI). The manuscript is very clear and pleasant to read and provides the reader with many references on this important topic including recently discovered marine antibiotics and anti-adhesives compounds.

To date, many antibiotics have been impregnated into the urethral catheters, including nitrofurazone, gentamicin, norfloxacin or vancomycin etc. The main concern with the present review is  that no marine product is actually used or even simply evaluated in CAUTI. So the authors took care to precise “potential new weapons ” in the title. It is indeed the case because, almost all references  cited in the present review  don’t  demonstrated any activity on the complex CAUTI but simple in vitro antimicrobial activity on standard bacterial strains . So, as it stands the review is quite strange, because if all the compounds listed have activity on some strains of bacteria involved in CAUTI, they might be used for any other infections related to a prosthetic or medical device.

Accordingly, to my opinion the manuscripts can clearly be published provide the authors emphasize much more clearly that it is a prospective proposal more than a review of accomplished works.  

Beside this main criticism, there are minor points to address.

Lines 213 and 248, replace esters and lactones by “polyketides”

Biosurfactant section. Most of these materials with few exceptions are unidentified mixtures of compounds (for example MSA19 or MSA10, lipopeptides of ref 147 etc). Further pharmaceutical development of such mixtures is really unlikely and the authors should emphasize this point.

Ref 124 seems unrelated to MAB36

Author Response

Point 1: Accordingly, to my opinion the manuscripts can clearly be published provide the authors emphasize much more clearly that it is a prospective proposal more than a review of accomplished works.  

Response 1: Revision is done. Please find the revisions in the ‘Abstract’ and section 6 ‘Opportunities, Challenges and Future Perspectives ‘.

Point 2: Lines 213 and 248, replace esters and lactones by “polyketides”

Response 2: Done. Please see the revisions in the attachment.

Point 3: Biosurfactant section. Most of these materials with few exceptions are unidentified mixtures of compounds (for example MSA19 or MSA10, lipopeptides of ref 147 etc). Further pharmaceutical development of such mixtures is really unlikely and the authors should emphasize this point.

Response 3: Please find the revisions in section 6 ‘Opportunities, Challenges and Future Perspectives ‘.

Point 4: Ref 124 seems unrelated to MAB36

Response 4: Revision is done. Please see updated ref 124.